# Evaluation of a Machine Learning Algorithm to Classify Ultrasonic Transducer Misalignment and Deployment Using TinyML

**DOI:** 10.3390/s24020560

**Published:** 2024-01-16

**Authors:** Des Brennan, Paul Galvin

**Affiliations:** Tyndall National Institute, University College, T12 K8AF Cork, Ireland; paul.galvin@tyndall.ie

**Keywords:** ultrasonic, machine learning, TinyML

## Abstract

The challenge for ultrasonic (US) power transfer systems, in implanted/wearable medical devices, is to determine when misalignment occurs (e.g., due to body motion) and apply directional correction accordingly. In this study, a number of machine learning algorithms were evaluated to classify US transducer misalignment, based on data signal transmissions between the transmitter and receiver. Over seven hundred US signals were acquired across a range of transducer misalignments. Signal envelopes and spectrograms were used to train and evaluate machine learning (ML) algorithms, classifying misalignment extent. The algorithms included an autoencoder, convolutional neural network (CNN) and neural network (NN). The best performing algorithm, was deployed onto a TinyML device for evaluation. Such systems exploit low power microcontrollers developed specifically around edge device applications, where algorithms were configured to run on low power, restricted memory systems. TensorFlow Lite and Edge Impulse, were used to deploy trained models onto the edge device, to classify signals according to transducer misalignment extent. TinyML deployment, demonstrated near real-time (<350 ms) signal classification achieving accuracies > 99%. This opens the possibility to apply such ML alignment algorithms to US arrays (capacitive micro-machined ultrasonic transducer (CMUT), piezoelectric micro-machined ultrasonic transducer (PMUT) devices) capable of beam-steering, significantly enhancing power delivery in implanted and body worn systems.

## 1. Introduction

Trans-tissue US power transfer has been demonstrated from shallow to deep implants [1], for battery recharge or powering devices directly. Compared to conventional electromagnetic energy transfer, US transducers offer reduced size, lower tissue attenuation, higher safe operating power (720 mWcm^−2^) [2] and the possibility for energy focusing (e.g., beam steering). With radio frequency (RF) charging, a limit of 1–10 mWcm^−2^ is placed by medical regulation authorities [3], thus US offers deeper signal delivery within safe operating power limits. Body motion is a significant challenge to US power transfer and minimal misalignment (<4 mm) has been shown to reduce acoustic energy by up 40% [4]. In US medical imaging, signal parameters (e.g., phase, timing) are optimised on transducer arrays for target depth focusing [5], similar approaches can be used to optimise implant power delivery. Many medical implants have communication capability, to exchange data with an external body mounted receiver to indicate device status or alter functionality.

An approach to guide the transducer alignment direction is required. Where implant systems exchange data signals, alignment may be achieved using the transmitted signal envelope, thus modifying transducer orientation to optimise implant power transfer. Envelope detection may already be implemented within such power transfer and communication systems. In US target location; (i) time of flight (TOF) [6], (ii) time difference on received US envelope [7] and (iii) envelope phase [8] have been demonstrated. The potential to use machine learning to classify US transducer alignment, using a communication signal between transmitter/receiver, has not previously been reported. ML has been used to automate complex system parameters, where algorithms are trained on a “train” dataset and independently evaluated on a “test” dataset. Training can be supervised (i.e., class labels pre-assigned to input training set) or unsupervised (i.e., the algorithm determines closely related inputs and labels them accordingly). In this work, ML algorithms were evaluated to classify misalignment between an US transmitter and receiver. The algorithms included auto-encoder, convolutional neural networks (CNN) implementing spectrogram analysis and a neural network (NN). Autoencoders and CNNs are extensively used in healthcare applications for complex time varying signal analysis, specifically around classifying heart arrhythmia [9] and respiratory sound analysis [10]. Where constituent frequency and intensity vary between normal and anomalous signals. Multilayer NNs are applied across a range of applications to identify hierarchical representations from complex inputs [11], without prior specialist training knowledge, thus specific signal preparation or parameter extraction steps are not required. These three algorithms can be applied directly to the US signals without the need for parameter extraction, as with conventional models such as decision trees, support vector machines (SVMs), K-means clustering and K-nearest neighbours. ML algorithm deployment on TinyML devices is still in its infancy, thus comparative performance metrics is difficult to find in the literature [12]. However, the algorithms considered here have previously been deployed on TinyML applications. An autoencoder was implemented [13] for urban noise recognition, while spectrogram analysis using CNN has been implemented for sound or word recognition [14]. The autoencoder also implements noise reduction through signal reconstruction and has been demonstrated with US communications [15]. The ability to learn non-linear relationships between input signals and outputs, allow NNs to model complex relationships within data. They implement a “black-box” type algorithm and it’s difficult to interpret how the algorithm arrives at a specific output, unlike conventional models (e.g., decision trees, support vector machines), where the decision making process is clearly defined (e.g., trees, using Entropy or Gini index). The best performing algorithm, was deployed onto a TinyML edge device for evaluation, where the final algorithm was optimised and deployed onto a local hardware system, referred to as an edge device [16,17]. Such systems typically exploit low power microcontrollers, developed specifically around edge device applications, where algorithms are configured to run on low power, restricted memory systems (e.g., Arduino NANO BLE 33). The benefits of TinyML applications include; (i) minimal data exchange with network servers, (ii) data stored locally on the device, (iii) no internet connectivity required, (iv) reduced model size, (v) low power consumption. The TinyML approach has been adapted by a number of system providers, with many developing AI platforms for application areas including; wearables, preventative maintenance, home and environment monitoring [18]. Such platforms are compatible with established ML open-source environments (e.g., Tensorflow, Keras, ONNX, SKlearn etc.), used in industry and research. Cloud service software platforms (e.g., AWS—Sagemaker Classic V1.5 [19], Microsoft—Azure V2.17 [20]) have also implemented edge capabilities offering the possibility to develop and deploy algorithms using compatible edge boards. Thus complex algorithms can be developed on cloud platforms and deployed as simplified models on edge devices.

In this work, TensorFlow Lite [21,22] was used to deploy a trained neural NN model onto an edge device to classify US communication signal envelopes, according to transducer misalignment extent. Such an approach could facilitate transducer realignment by the user or an autonomous system. In medical devices, high US frequencies (100′s kHz to MHz range) are typically used for power delivery, requiring expensive instrumentation (e.g., Verasonics [23]). In this work, to readily generate a large dataset for algorithm development, we used low cost off the shelf transducer systems operating at 40 kHz in air, as misaligned signal behaviour was similar to high frequency systems (2–10 MHz), operating on gel phantoms.

## 2. Materials and Methods

### 2.1. US Signal Acquisition

Two US transmitter modules (JSNSR-04T) were used as transmission/receiver trans-ducers (Figure 1a). An Arduino Uno, powered and controlled the transmitter/receiver components. Such a system was previously demonstrated in underwater, low power US data transmission over short distances [24]. The US transducer had a peak transmission at 40 kHz with 3 kHz bandwidth. The transmitter was triggered by a 10 µs TTL pulse, subsequently emitting an eight cycle pulse at 40 kHz, this signal was sampled by the receiving transducer and simultaneously recorded on a digital oscilloscope (R&S^®^RTO2000). The horizontal transmitter displacement, relative to the receiver was varied from 0 mm to 20 mm as illustrated in Figure 1b and signals recorded across eight positions. For the classification algorithm, each position was given a label; position 0 = aligned, position 1 = misalligned 1, position 2 = misalligned 2, position 3 = misalligned 3, position 4 = misalligned 4, position 5 = misalligned 5, position 6 = misalligned 6, position 7 = misalligned 7.

The data set consisted of 736 recorded signals, transmitted between two US transducers at varying extent of horizontal transducer misalignment (0–20 mm). Similar numbers of data files for each class were sampled for receiver displacements, achieving a balanced dataset. A typical US signal exchanged between the transmitter and receiver modules is illustrated in Figure 1c, i.e., the analogue signal received at the transducer. Each signal was assigned a label to classify the extent of transducer misalignment, upon which the algorithms were trained and tested.

### 2.2. Data Pre-Processing

Each signal file (.csv format) recorded on the oscilloscope was loaded into a Python dataframe structure, for data processing and algorithm model input. For the autoencoder and NN, the input data was the positive signal envelope as illustrated in Figure 2b. This was extracted from raw signals (Figure 2a) using the Hilbert transform function in Python. This step was implemented for each data file, forming a dataset composed entirely of positive signal envelopes. Data was configured as an X matrix with each row representing a signal transmission, while the labels associated with each signal alignment were represented by a vector y. The X matrix was composed of continuous numerical data (i.e., signal envelope amplitude), which was normalised over the range 0–1. The dataset was considered balanced with a similar number of data signals in each label category. The X, y data structures were subdivided into training and test subsets, with a 70:30 split. Before the data split, X and y were randomly shuffled and maintained the same class ratios in test/train datasets as the original dataset.

For CNN classification, the original raw data (Figure 2a) was used to generate signal spectrogram (e.g., Figure 3), forming a second dataset. A python function was used to generate spectrograms for each original US signal. In Figure 3, the spectrogram *X*-axis was time (seconds) while the *y*-axis was frequency. The intensity of the image color map illustrated specific frequencies present within the signal time window. The 40 kHz signal associated with the peak US transducer operating frequency, was evident over a duration of approximately 1.5 ms.

In this case, X_test and X_train datasets were formed by allocating spectrograms to specific class folders, each labelled according to signal misalignment. Spectrograms were randomly assigned to train and test datasets with similar numbers assigned to each class folder, maintaining balanced datasets.

### 2.3. ML Algorithms

The data subset X_train, y_train was used to train algorithms, while the data subset X_test, y_test was used to investigate algorithm performance. Algorithm performance used metrics based on the number of true positives (TP), false positives (FP), true negatives (TN) and false negatives (FN) returned by the algorithm. Metrics included;
Accuracy—ratio of correct predictions from all predictions:
$$Accuracy=\frac{TP+TN}{TP+TN+FP+FN}$$Precision—proportion of correct class predictions over all positive predictions:
$$Precision=\frac{TP}{TP+FP}$$Recall—proportion of actual positives identified correctly:
$$Recall=\frac{TP}{TP+FN}$$

Performance metrics were generated using Tensorflow and Keras libraries implemented in python for classification statistics, confusion matrix etc. The algorithms were suitable for numerical, categorical and nominal data parameters, but only numerical variables were used in the X matrix and the y vector was categorical using one-hot-encoding.

#### 2.3.1. Autoencoder

The autoencoder was composed of a neural network forming two blocks, (i) the encoder (Figure 4a) and (ii) the decoder (Figure 4b), with the former realising data reduction and the latter reconstructing the input signal as close to its original state as possible. With data reduction, relevant signal parameters were preserved while non-essential components were removed.

The encoder input was the signal envelope vector (301 × 1) which passed through three layers, of node sizes 32, 16 and 8, with the latter forming the encoded output latent representation of the input signal. The decoder input was this latent representation vector, which was used to reconstruct the input signal by passing it through sequential NN layers of size 16, 32 and 301. Thus the output layer had the same dimension as the input layer, effectively reconstructing the input signal. During training the NN node weights were modified based on the adaptive moment estimate (ADAM) optimiser to minimise the training error, based on mean square error (MSE). ADAM uses second and third moments to estimate loss and direct gradient descent, optimising hyper-parameters for all nodes. The autoencoder was a one versus all algorithm, trained to identify a target signal (referred to as normal) and distinguish from remaining signals (referred to as anomalous). The algorithm was trained over 1000 epochs using batch data size of 512 and it was found the loss on the validation dataset for normal signal had a plateau at 0.06, while the anomalous signal data achieved a loss of 0.08 after 300 epochs. It was also noted from Figure 4c how training loss continued to decrease with increasing epoch number, thus an independent test dataset gave a non-biased performance loss estimate and was more representative of likely algorithm performance on unseen data. The algorithm reached optimum parameter settings on test data at 400 epochs, thus a call-back function was used to identify these settings in the final model. The algorithm determined an error threshold between the reconstructed signal and the original input signal, above this threshold the signal was classified as anomalous, while signals below threshold were classified as normal. The threshold was based on the mean error on the normal dataset plus N standard deviations (1.0), varying N modified the threshold and thus algorithm performance, in this work N = 3.
threshold = mean (train_loss) + N.std (train_loss) (1.0)

Modifying the threshold parameter N changed the FP, FN density in model results and hence performance metrics (precision, accuracy, recall). Thus N was optimised to achieve acceptable performance for a specific application, i.e., reduce FP or FN in the train dataset results. In some applications (e.g., medical diagnosis) it may be acceptable to have increased numbers of FP over FN, as missing a diagnosis has significant negative impact.

#### 2.3.2. Convolutional Neural Networks (CNNs)

CNNs (Figure 5a) was the second classification algorithm evaluated, this was applied to spectrograms generated from original US signals. The spectrograms formed input to the CNN, which extracted key frequency domain features.

Spectrograms were generated in Python forming eight classes within test and train data sets. The batch size was set to thirty-two. The CNN model was defined using sequential structures from the Tensorflow and Keras platforms. The CNN classifier was composed of three Conv2D layers each of which acted as convolutional filter to extract features and map a reduced representation to the following CNN layer. The Relu activation threshold removed negative values after each convolution layer. The maxpooling layer was an n × n sliding filter which extracted key features from each filter step, across the spectrogram. Maxpooling reduced spectrogram size/representation from proceeding to follow-on convolutional layers. It also reduced the number of connected nodes, while maintaining key frequency feature information. The maxpooled 2D feature maps were flattened into a single vector, each node in this output vector represented a specific spectrogram feature. In the final model, two dropout layers were included (0.25, 0.5) to remove any low importance nodes from dense node layers, this reduced complexity and shortened training time. Two of the CNN optimisation hyper-parameters included the number of layers within the neural network and layer density. Additional layers of varying density were added to the network to determine if algorithm performance improved. The model was compiled using ADAM optimisation on a cross entropy loss function. The training and validation datasets were evaluated over 800 epochs (Figure 5b) with a call-back function identifying the optimum model for evaluation. The train/validation losses and accuracy (Figure 5b) reach a plateau after approximately 200 epochs, with the validation dataset performance more representative of expected performance in the final CNN classifier model, at approximately 80% accuracy, with a loss based on cross entropy of approximately 0.5. The oscillations in the validation loss plotted over epochs, result from smaller batch sizes repeatedly used in the validation set over epochs, thus some data combinations gave better algorithm performance than others. This was also evident in algorithm performance evaluation, where some transducer positions achieved better performance metrics over others.

#### 2.3.3. Neural Network (NN)

The NN had an input layer, output layer and a number of hidden layers defining its structure. The layer number and nodes per layer were hyper-parameters optimised during the training process. Since eight classes were considered, the output layer was a vector of dimension 8 × 1 (i.e., one node representing each class), with a softmax activation function determining which output node was activated to classify the input signal. The data used in this NN algorithm was the positive envelope extracted from the US signal transmitted between the transducers. The model was trained by modifying node weights during feed-forward/feedback adjustment, implemented by the optimiser function on the loss metric. The ADAM optimiser was used on the cross entropy loss function. The complete envelope dataset was randomly split into test and train datasets, with 70:30 ratios. Next the test and train datasets were normalised in the range 0–1. The structure of the network (Figure 6a) was; (i) the input layer (301 × 1), (ii) four dense layers (128 × 1) and (iii) the softmax output layer (8 × 1, Relu activation). After the model was compiled, it was trained over 100 epochs with the accuracy and loss metrics evaluated during this step (Figure 6b).

A probability based prediction function was generated from the model using the Softmax function and evaluated on the test dataset. For each test signal a probability vector (8 × 1) was calculated, the maximum probability position represented the most likely class, according to the NN classifier model. A recall function was used to identify the best model parameters, which were used in the final model.

### 2.4. Edge Device Algorithm Deployment

Algorithm development was undertaken in Python using TensorFlow (TF) and KERAS, with the former facilitating deployment on mobile, embedded and edge devices, using TF-Lite. TF-Lite executes a low resource algorithm version on devices with limited memory and compute power. The full TF model was easily converted into a TF-Lite format by running conversion scripts [25]. The resulting TF-Lite model had reduced size and complexity in moving from 32-bit to 8-bit numerical representation, using dropout layers and removing features with minimal impact on algorithm performance. The TFLite model was stored as a FlatBuffer [26], which was useful reading large data chunks one piece at a time, rather than load everything into RAM. TF-Lite models can be deployed on; (i) Android and ios devices, (ii) embedded Linux using Raspberry Pi, Coral devices, (iii) Microcontrollers etc.

The final algorithm was deployed and evaluated on a TF-Lite supported development microcontroller board, the Arduino Nano 33 BLE Sense. This board was chosen because of low cost, availability and ease of ML algorithm implementation. For model deployment, the TF-Lite algorithm model library was downloaded onto the Nano BLE 33 board via the Arduino IDE using the “manage libraries” tab [27]. The algorithm was compiled and uploaded directly onto Arduino memory via the serial port. Fast ADC settings were implemented on the edge device to optimise data acquisition by manipulating ADC buffers (ADPS0, ADPS1, ADPS2, ADCSRA). The receiver signal could also be down shifted for edge board audio input. The steps for data preparation, model training and evaluation, followed by edge device deployment are outlined in Figure 7.

The edge impulse platform could also be used to build an algorithm project which could be deployed directly onto the edge device [28]. The serial monitor was also used to visualise in real time, the algorithm inference running on the board based on the transmitted US signal. The inference output was the probability the input signal belonged to a specific transmitter transducer position class. For a specific alignment signal, the algorithm output was a probability between 0 and 1, a value close to 1 indicated high probability of belonging to that position class. The serial interface displayed the neural network classifier model probability assigned to class labels, representing the extent of transducer misalignment, as illustrated in Figure 7 “model inference output”.

## 3. Results

### 3.1. Algorithm Performance Evaluation

The classification algorithms were compared using; accuracy, precision, recall, true positives, true negatives, false positives and false negatives. Such metrics are typically used to evaluate and compare algorithm classification performance. Different thresholds were also considered to determine the best performance metrics. Envelope signals from the aligned class (position 0) were input to the trained autoencoder model, Figure 8 highlights the original input signal (blue) and the decoder reconstituted signal (red), the error between the two signals was also plotted (red fill). In the case where error was below threshold (1.0), as with signals in Figure 8a,b, these were correctly classified as aligned (position—0) by the autoencoder algorithm. Where signals from transducer positions misaligned 4 & 7 were input to the algorithm, a high error between input and reconstructed signal resulted, as illustrated in Figure 8c,d.

The autoencoder performance for each US transducer configuration was outlined in Table 1, where labels “Misalign-1” to “Misalign-7” represent the extent of transducer misalignment. Good accuracy, precision and recall was achieved for class labels misalign-2, misalign-3, misalign-5 and misalign-7. Precision and accuracy was good for misalign-1, misalign-4 and misalign-6, however recall was poor. For the aligned class, precision and recall was poor due to increased numbers of FP and FN.

The CNN spectrogram classifier performance was also evaluated using accuracy, precision and recall on the test data set (Table 2). The classifier was evaluated with a number of CNN configurations, including layer depth and node number. The best model performance was achieved with 2D convolution layers set to densities 16, 12, 8 respectively, configurations with 64, 32, 16 layers achieved no improvement in performance.

Classification performance was good for four transducer positions (aligned, Misalign-1, Misalign-3, Misalign-7), while Misalign-5 and Misalign-6 had good accuracy and precision, but low recall. Poor precision and recall were associated with classes misalign-2, misalign-4. For such transducer positions, the algorithm confused classes with positions in close proximity and misclassified accordingly.

The performance of the NN model with 4 hidden layers, each with 128 nodes was comparable to the models evaluated with 5 & 6 hidden layers, thus the simpler network configuration was preferred over deeper models. This simple NN model (Table 3) performed superior to the autoencoder and spectrogram classifiers. The accuracy, precision and recall metrics were consistently above 0.88 for all alignment positions. The average model accuracy, precision and recall as outlined in Table 4, also indicated better NN model performance. Thus the NN model was selected for deployment onto the edge device evaluation board.

### 3.2. Edge Device Performance

The NN algorithm was deployed onto the edge devices to classify US signals transmitted between transducers. This offered the possibility to implement in near real time, position inference based on US signal, as illustrated in Figure 9. The neural network classifier model output assigned a probability to class labels, representing the extent of transducer misalignment.

From the classifier probability output outlined in Table 5, all positions are clearly labelled correctly, with high probability for each transducer position. As the US transmitter transitioned between labelled positions, the inference probability gradually reduced for the departing position label and increased for the arriving position label. Thus midway between the two positions, probability approached 0.5 for adjacent position labels. The coloured rows in Table 5, correspond to transducer positions outlined in Figure 9a–c. The algorithm implementation time on the evaluation board typically took 319 ms. Moving from 32 bit float to 8 bit integer model representation, reduced required memory from 34.4 kb to 27.7 kb.

It was also possible to use the Edge Impulse platform as a remote server upon which classification was implemented. Thus the board locally acquired data for transmission to the Edge Impulse server, for “cloud” based classification. After signal analysis the alignment class was conveyed back to the local pc running the Edge Impulse platform. From signal acquisition to locally display the algorithm output class, took approximately 10 s. Thus there was significantly increased latency in cloud implementation compared to local on device algorithm application. The second issue arising with cloud based implementation was connectivity, poor broadband availability or loss of signal disrupting algorithm inference. The server sites experiencing technical difficulties or upgrading systems also impacted device connectivity and inference continuity.

## 4. Discussion

A data set based on a transmitted 8-bit US signal between a transmitter and receiver, across a range of displacements, was used to access the ability of three algorithms classifying transducer misalignment. The models included an autoencoder, multilayer NN and CNN classifier. The data set was composed of 736 signals acquired across eight positions, with a similar number of samples per class. For each model, performance relevant hyper-parameters were optimised based on a selected optimiser (e.g., ADAM), minimising loss (e.g., cross entropy) over a range of epochs. The NN model achieved the best classification performance metrics based on accuracy, precision, recall and confusion matrices, across transducer displacements. The autoencoder was more accurate for misaligned than aligned classes and envelope shape similarity for shallow misalignment angles may be an issue. CNN classifier applied to data spectrograms didn’t perform as well as the other algorithms. The NN model delivered the best performance and was implemented on an edge device, the NanoBLE33. The algorithm successfully classified misalignment between an ultrasonic transmitter and receiver, thus demonstrated a potential approach to transducer alignment based on a short US data exchange. This result is relevant to transducer alignment in US power transfer or communication applications, especially for arrayed US devices (e.g., CMUT, PMUT), where altering beam phase and intensity can steer signals to specific tissue locations. The Edge Impulse software platform offered a convenient approach for algorithm deployment on supported edge devices. The Tensorflow model was imported into the Edge Impulse environment where code was generated and burnt directly onto hardware. In deploying the algorithm an Int8 TFLITE model representation was selected to minimise memory requirements. ML signal classification took typically <350 ms, an acceptable time period for this alignment evaluation. The benefit of edge ML deployment over cloud based application was clearly demonstrated with reduced inference time (milliseconds vs. seconds) and minimal disruption due to connectivity or remote server issues. With more complex algorithms higher computational power and memory may be required, which may not be available on an edge device. Thus cloud servers may be required. Cloud based systems also facilitate augmented/continuous algorithm training to improve accuracy and robustness signal variation (e.g., drift). Additional consideration such as data integrity, update rate, hardware size and power consumption, should be considered when deciding where and how ML applications are deployed.

## Figures and Tables

**Figure 1 sensors-24-00560-f001:**
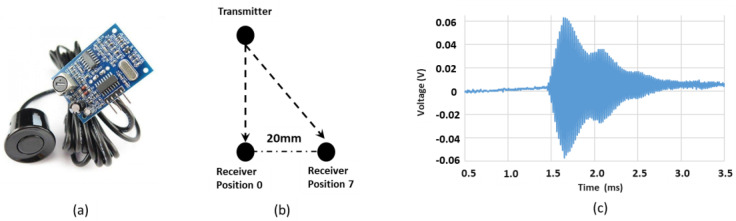
(**a**) The US transducer module used to transmit data between transducers; (**b**) receiver module displaced across a range of horizontal misalignment positions, relative to transmitter (indicated by arrows) (**c**) example of recorded US signal (aligned—position 0).

**Figure 2 sensors-24-00560-f002:**
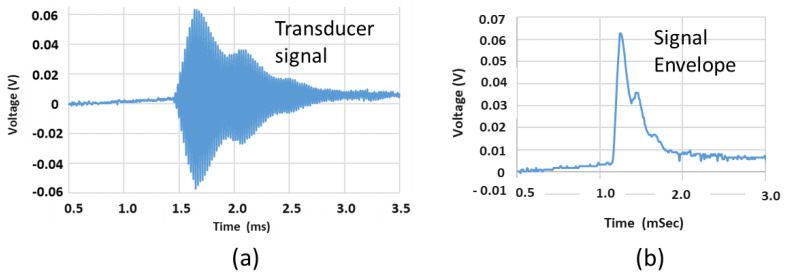
(**a**) The raw US signal acquired on the receiver transducer and sampled by an oscilloscope; (**b**) the positive signal envelope extracted using a Hilbert transform prior to algorithm implementation.

**Figure 3 sensors-24-00560-f003:**
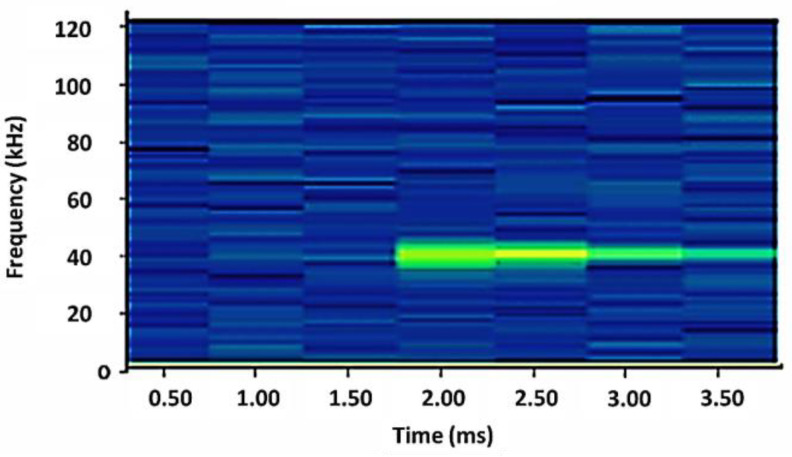
An example of the spectrogram color map extracted from an unprocessed US signal, transmitted between transducers in the aligned configuration. The 40 kHz signal (green) is clearly defined against background frequencies (blue).

**Figure 4 sensors-24-00560-f004:**
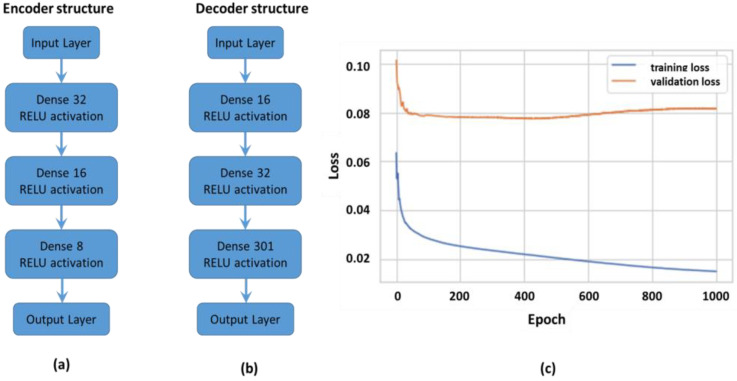
(**a**) The NN structure for the encoder (arrow indicates algorithm sequence); (**b**) the decoder, implemented in this work (arrow indicates algorithm sequence); (**c**) the NN model loss over training epochs.

**Figure 5 sensors-24-00560-f005:**
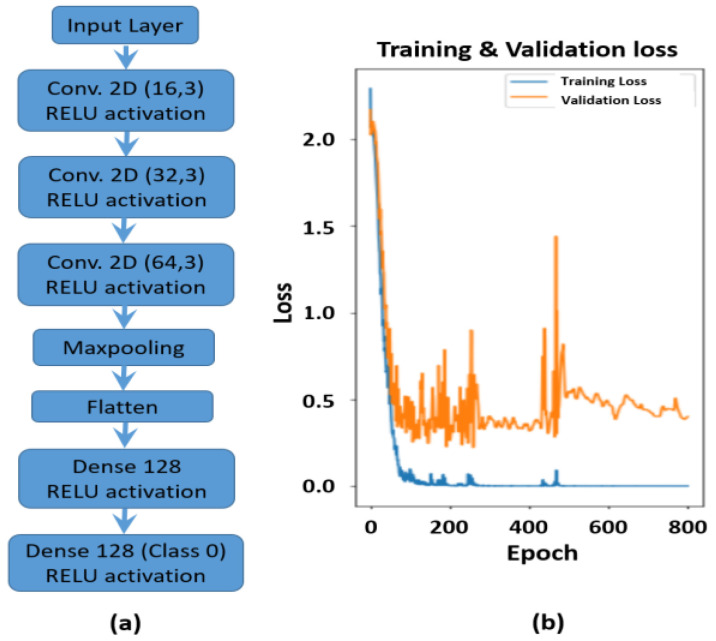
(**a**) The CNN structure implemented for spectrogram analysis (arrow indicates algorithm sequence); (**b**) the algorithm training loss over training epochs.

**Figure 6 sensors-24-00560-f006:**
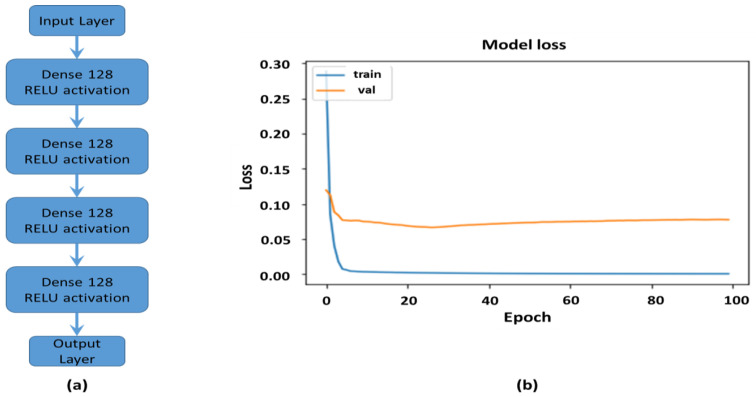
(**a**) The implemented NN structure (arrow indicates algorithm sequence); (**b**) and the algorithm training loss over training epochs.

**Figure 7 sensors-24-00560-f007:**
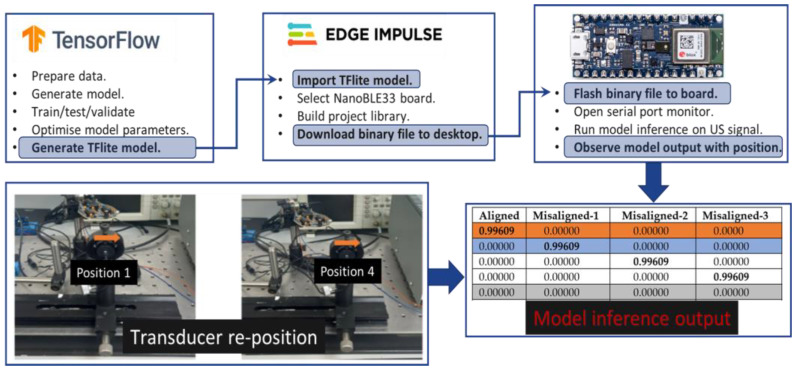
Outline of data preparation and model development in TensorFlow (Python) to deployment on board using the edge impulse platform and model inference output as transducer position changes (

 position 0, 
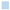
 position 1, 
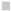
 position 4).

**Figure 8 sensors-24-00560-f008:**
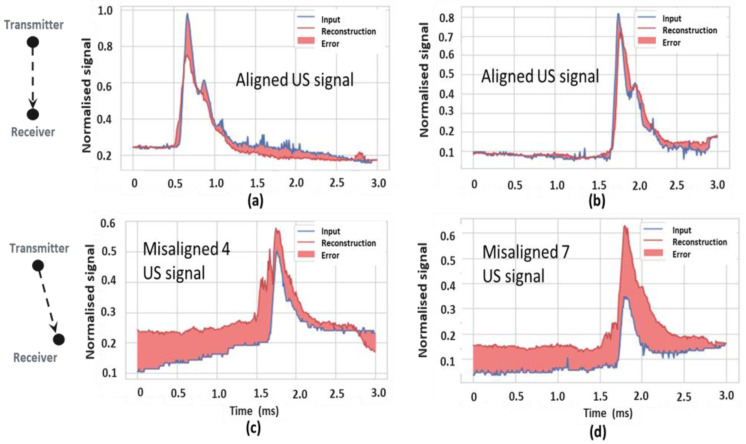
(**a**,**b**) Original and reconstructed signals for aligned input; (**c**,**d**) high error was evident for reconstructed input signal envelopes, misaligned 4 and 7.

**Figure 9 sensors-24-00560-f009:**
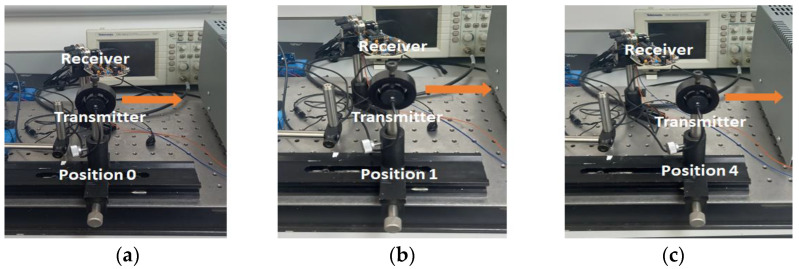
Outline of three transmitter/receiver alignment configurations, classified by edge deployed algorithm; (**a**) position 0 (aligned); (**b**) misaligned position 1; (**c**) misaligned position 4, where the arrow indicates direction of transmitter movement.

**Table 1 sensors-24-00560-t001:** Performance metrics for the auto-encoder classification algorithm outlined for aligned and misaligned US signals.

Configuration	Accuracy	Precision	Recall	TP	TN	FP	FN
Aligned	0.87	0.22	0.41	6	187	21	7
Misalign-1	0.92	0.90	0.56	18	187	2	14
Misalign-2	0.92	0.72	0.79	27	177	10	7
Misalign-3	0.96	0.85	0.78	18	195	3	5
Misalign-4	0.91	0.73	0.56	7	185	13	6
Misalign-5	0.95	0.87	0.84	27	185	4	5
Misalign-6	0.93	0.82	0.53	14	192	3	12
Misalign-7	0.94	0.88	0.70	22	187	3	9

**Table 2 sensors-24-00560-t002:** Illustrated how four classes perform well with the CNN algorithm (Aligned, Misalign-1,3,7), while the others have mixed performance.

Configuration	Accuracy	Precision	Recall	TP	TN	FP	FN
Aligned	1.00	1.00	1.00	3	47	0	0
Misalign-1	0.96	1.00	0.78	7	41	0	2
Misalign-2	0.94	0.75	0.60	3	44	1	2
Misalign-3	1.00	1.00	1.00	4	46	0	0
Misalign-4	0.86	0.25	0.66	2	41	6	1
Misalign-5	0.82	1.00	0.36	5	36	0	9
Misalign-6	0.92	1.00	0.56	5	41	0	4
Misalign-7	0.98	1.00	0.88	8	41	0	1

**Table 3 sensors-24-00560-t003:** The NN model was evaluated for layer depth 4, using 128 nodes on hidden layers.

Configuration	Accuracy	Precision	Recall	TP	TN	FP	FN
Aligned	0.97	1.00	0.77	10	208	0	3
Misalign-1	0.97	0.91	0.88	29	185	3	4
Misalign-2	0.98	0.94	0.91	32	184	2	3
Misalign-3	1.00	1.00	1.00	23	198	0	0
Misalign-4	0.99	0.96	1.00	30	190	1	0
Misalign-5	0.98	0.92	0.89	23	193	2	3
Misalign-6	0.98	0.93	0.90	26	190	2	3
Misalign-7	0.98	0.96	0.90	27	190	1	3

**Table 4 sensors-24-00560-t004:** The average accuracy, precision and recall of the three algorithms evaluated on the US alignment dataset.

Model	Avg. Accuracy	Avg. Precision	Avg. Recall
Autoencoder	0.93	0.75	0.65
Spectrogram-CNN	0.94	0.88	0.73
Neural network	0.98	0.95	0.91

**Table 5 sensors-24-00560-t005:** Outline of the probability for transducer misalignment for positions 0–7, the high probability score returned by the on-board algorithm correctly classified the alignment position for the edge device board (

 position 0, 
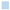
 position 1, 
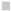
 position 4).

Aligned	Misaligned-1	Misaligned-2	Misaligned-3	Misaligned-4	Misaligned-5	Misaligned-6	Misaligned-7
0.99609	0.00000	0.00000	0.0000	0.00000	0.00000	0.00000	0.00000
0.00000	0.99609	0.00000	0.00000	0.00000	0.00000	0.00000	0.00000
0.00000	0.00000	0.99609	0.00000	0.00000	0.00000	0.00000	0.00000
0.00000	0.00000	0.00000	0.99609	0.00000	0.00000	0.00000	0.00000
0.00000	0.00000	0.00000	0.00000	0.99609	0.00000	0.00000	0.00000
0.00000	0.00000	0.00000	0.00000	0.00000	0.99609	0.00000	0.00000
0.00000	0.00000	0.00000	0.00000	0.00000	0.00000	0.99609	0.00000
0.00000	0.00000	0.00000	0.00000	0.00000	0.00000	0.00000	0.99609

## Data Availability

The data presented in this study are available on request from the corresponding author.

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
