# Peer review of "Evaluation of a Machine Learning Algorithm to Classify Ultrasonic Transducer Misalignment and Deployment Using TinyML"

_sensors, 2024, doi:10.3390/s24020560_

Round 1

Reviewer 1 Report

Comments and Suggestions for Authors

In this manuscript, the authors propose a neural network approach to classify ultrasonic transducer misalignment. Overall, this work has certain innovation and high practical value. However, certain issues need to be addressed before considering it for publication.

1. Table 1 and Table 4 should not be represented by graphs.

2. The figures in your paper are a bit blurry, such as Fig 1.c, Fig 3, Fig 4..., and some pictures were flattened, such as Fig 5, Fig 7. Please consider replacing them with clearer ones.

3. The expression of numbers and units in the text is not standardized enough.

4. Contribution points need to be rephrased. These points contain minute-level details and contributions.

5. Some abbreviations should be placed where they first appear in the text. Line 334.

6. If possible, the authors of the study can present the justification for choosing the specific models they worked on comparatively.

Author Response

We'd like to thank the reviewer for reviewing the submitted manuscript and the authors responses are in the attached document.

Reviewer 2 Report

Comments and Suggestions for Authors

In this study, a number of machine learning algorithms were evaluated to classify US transducer misalignment, based on data signal transmissions between transmitter and receiver. Over seven hundred US signals were acquired across a range of transducer misalignments. Signal envelopes and spectrograms were used to train and evaluate machine learning (ML) algorithms, classifying misalignment extent. The algorithms included auto-encoder, convolutional neural networks (CNNs) and neural networks (NNs). The best performing algorithm, was deployed onto a TinyML edge device for evaluation. TensorFlow Lite and Edge Impulse, were used to deploy trained models onto the edge device, to classify signals according to transducer misalignment extent. TinyML deployment, demonstrated near real-time (< 350msec) signal classification achieving accuracies > 99%. This opens the possibility to apply such ML alignment algorithms to US arrays (capacitive micro-machined ultrasonic transducer - CMUT, piezoelectric micro-machined ultrasonic transducer - PMUT devices) capable of beam-steering, significantly enhancing power delivery in implanted and body worn systems.

This study provides novel and important results. In addition, it is very well organized. Meanwhile, the authors should consider the following minor issue:

-There is need for a proofreading of the paper. Afterwards, it can be accepted for publication.  

Comments on the Quality of English Language

There is need for a proofreading of the paper. Afterwards, it can be accepted for publication.  

Author Response

(The authors gave the same response as above.)

Reviewer 3 Report

Comments and Suggestions for Authors

The manuscript compares the performance of different ML algorithms for accurately predicting the misalignment positions and their deployment on an edge device application. The topic is of high interest, and the authors have adequately explained the study, its contributions, the related literature, and future work. I think the manuscript is acceptable for publication in the sensors journal.

The manuscript compares the performance of different ML algorithms for accurately predicting the misalignment positions and their deployment on an edge device application. The authors have adequately explained the study, its contributions, and the related literature. I think the manuscript is acceptable for publication in the sensors journal.

I have one concern- why do the input waveforms in Figures 8a and 8b look so different if both have the same positions (aligned)? Have you quantified the repeatability of the data acquisition? 

Author Response

(The authors gave the same response as above.)
